# Piezoelectric Energy Harvesting Solutions: A Review

**DOI:** 10.3390/s20123512

**Published:** 2020-06-21

**Authors:** Corina Covaci, Aurel Gontean

**Affiliations:** Applied Electronics Department, Politehnica University Timisoara, 300006 Timisoara, Romania; aurel.gontean@upt.ro

**Keywords:** energy harvesting, piezoelectric materials, piezoelectric transducer types, modeling, frequency response, energy harvesting electronic circuits, SPICE simulation

## Abstract

The goal of this paper is to review current methods of energy harvesting, while focusing on piezoelectric energy harvesting. The piezoelectric energy harvesting technique is based on the materials’ property of generating an electric field when a mechanical force is applied. This phenomenon is known as the direct piezoelectric effect. Piezoelectric transducers can be of different shapes and materials, making them suitable for a multitude of applications. To optimize the use of piezoelectric devices in applications, a model is needed to observe the behavior in the time and frequency domain. In addition to different aspects of piezoelectric modeling, this paper also presents several circuits used to maximize the energy harvested.

## 1. Introduction

In recent decades, wireless technologies and microelectronics have led to the development of wearable devices, such as items of clothing and accessories, in which power is supplied either by batteries or energy harvesting devices [1]. In conjunction with these approaches is the concept of the Internet of Things (IoT), where wireless sensors networks are commonly used [2]. IoT has led to smart equipment being placed in remote areas or in places where it can be difficult or impossible to charge batteries (e.g., health care devices placed inside the human body, and smart buildings). Despite the progress made by low-power integrated circuit technology, the energy density of chemical batteries needs to be improved, since the power requirement for the mentioned applications is difficult to fulfil [3,4]. Therefore, it is necessary to develop new energy harvesting techniques to sustain such self-powered systems. Not only is energy harvesting in sustaining self-powered systems necessary as a feasible and economically practical alternative to batteries, it also reduces the danger of greenhouse gas emission and sustains the environment [5]. Typically, an energy harvesting system has three parts [6]:The energy source: represents the energy from which the electrical power will be scavenged—this energy can be ambient (available in the ambient environment, e.g., sunlight, ambient heat or wind) or external (energy sources that are explicitly deployed, e.g., lightning, human heat or vibrations) [7];The harvesting mechanism: consists of the structure which converts the ambient energy into electrical energy;The load: the sink which consumes or stores the electrical output energy.

The most common small-scale energy sources are sunlight, electromagnetic radiation, environmental mechanical energy, human body heat, and human body mechanical energy. Unlike solar energy, electromagnetic radiation, and environmental mechanical energy, which are highly dependent on the environment, human body energy harvesters can be integrated into daily human activities to power a variety of devices [3,8].

Human body heat can be harvested using the principle of thermoelectric power generators, based on the Seebeck effect of materials. Using this material’s principle, one can generate electrical energy using the difference between the human body and the ambient temperature. The inconvenience is that a considerable difference in temperature is needed to have a stable system [3].

Human body mechanical energy and environmental mechanical energy are widely exploited due to their abundance in daily life. Every motion in nature can be a potential source of kinetic energy. Therefore, mechanical energy is the most prevalent form of energy [9]. Using mechanical energy scavenging, sufficient power can be provided to ensure long-term autonomy for self-powered systems [1]. For example, around 10 mW can be harvested from the motion of the upper limbs, 1 mW can be obtained from a typing motion, breathing generates around 100 mW and, by walking, we generate up to 1 W [10]. The harvested power density $P_{res}$ for mechanical energy depends on the motion frequency and magnitude, as shown in the resonance power Formula (1) [11]:
(1)$$P_{res}=4\pi^{3}mf_{res}^{3}yZ_{max}$$
where m is the inertial mass, $Z_{max}$ is the maximum displacement, $f_{res}$ is the resonance frequency, and y is the amplitude of vibration of the housing.

In Figure 1 [9], the approximate working-frequency level for different mechanical energy sources is shown.

Energy harvesting efficiency can be defined as the ratio between the power consumed on the external load resistance and the total input mechanical power. Mechanical efficiency is the converted electrical power divided by the total input mechanical power, and the electrical efficiency can be defined as the ratio between the power consumed on the external load resistance and the converted electrical power [12]. Mechanical energy $E_{m}$ (Equation (2)), electrical energy $E_{e}$ (Equation (3)) and energy conversion efficiency $E_{\%}$ (Equation (4)) are defined using the following equations [13]:(2)$$E_{m}=\int_{0}^{\Delta t}F\;d\left(t\right)dt$$
(3)$$E_{e}=P\;\Delta t=\frac{V^{2}}{R}\;\Delta t$$
(4)$$E_{\%}=\frac{E_{e}}{E_{m}}\;100$$
where F is the applied force, d is the movement distance while the force is applied, Δt is the generation time, P is the output power, V is the output voltage, and R is the resistive load applied to the harvester.

A mechanical energy harvester can be used to harvest the energy generated by human walking. For this, there are two types of mechanical energy storage devices: flywheels (which have high energy densities, but also require a considerable amount of space and have complex structures) and springs (which have a low energy density, but are simple and reliable) [3]. Combining these harvesters with a ratchet, the mechanical harvested energy can be stored for later use.

There are three typical ways to convert mechanical energy into electrical energy: electromagnetic, electrostatic/triboelectric, and piezoelectric [14]. If high efficiency in the mechanical-to-electrical energy transfer is required, electromagnetic systems are the most suitable because they usually involve coils and magnets, but this also means bulky and complicated mechanisms [15]. The choice between the three methods is highly dependent on the application but, of these, piezoelectricity is the most widely studied [15].

If the application needs high voltage, high energy density, high capacitance, and little mechanical damping [16,17,18], piezoelectric energy harvesting is the solution, with the observation that piezoelectric materials can be brittle or rigid and can be toxic [19,20]. For applications that do not need outside sources, instead of piezoelectric transducers, electromagnetic devices can also be used [17]. They have high output current, low output impedance, and do not need contacts [18,21], but they usually have coil losses, low efficiency at low frequencies, and low output voltages [18,22]. Due to the power requirement, speed growth gears are added to meet desirable rotating velocity. Therefore, electromagnetic devices are not compatible with fabrication at the microscale level suitable for human body applications [23]. Triboelectric energy harvesting presents a multitude of advantages compared with piezoelectric and electromagnetic energy harvesting, such as high power density, high conversion, and device flexibility [23,24]. Still, it also has reliability and durability issues, and, in addition, its working mechanism is not yet fully understood [23,25].

Each energy harvesting method has advantages and disadvantages, and different approaches for harvesting energy effectively from human body motion are proposed in the literature.

Techniques to increase efficiency for piezoelectric energy harvesting include:Non linearity [26,27];Double pendulum system [28];Frequency up conversion [29];Circuit management [10,30].

Techniques to increase efficiency for electromagnetic energy harvesting include:Sprung eccentric rotor [31];Frequency up conversion [32];Spring clockwork mechanism [33];Spring-less system [34];Non linearity [35].

Techniques to increase efficiency for triboelectric energy harvesting include:Ultrathin flexible single-electrode [36];Core-shell structure mechanism [37];Air-cushion mechanism [38];Liquid metal electrode [39].

In Figure 2, the classification of energy harvesting sources is represented.

For self-powered systems, small-scale energy harvesters are ideal due to their advantages of small volumes, long lives, and low or non-existent need of maintenance [40].

In this review, we will focus on piezoelectricity and on the methods to harvest piezoelectric energy. The paper is organized as follows: in Section 2, a brief introduction to piezoelectricity is presented, followed by a classification of piezoelectric materials (Section 3) and a small piezoelectric transducers description (Section 4). Section 5 presents the parameters that need to be highlighted in piezoelectric modeling, while Section 6 shows the frequency response of a piezoelectric energy harvester. After the behavior of piezoelectric transducers is known, different electronic circuits for energy harvesting are presented in Section 7. The paper ends with a brief mention of some applications from the literature, some personal contributions, and conclusions.

## 2. Piezoelectricity

Briscoe and Dunn [41] defined piezoelectricity as “electric charge that accumulates in response to applied mechanical stress in materials that have non-centrosymmetric crystal structures”, while Erturk and Inman [42] defined piezoelectricity as “a form of coupling between the mechanical and electrical behaviors of ceramics and crystals belonging to certain classes”. The Greek origin of the word “piezoelectricity” is “squeeze or press” [5], which refers to the propriety of the piezoelectric materials to generate an electric field when a mechanical force is applied, a phenomenon called the direct piezoelectric effect [43].

The piezoelectric effect is divided into two phenomena: the direct piezoelectric effect and the converse piezoelectric effect [42]. The property of some materials to generate an electric field when a strain is applied (direct piezoelectric effect) was discovered by Pierre and Jacques Curie in 1880 [5]. The converse or inverse piezoelectric effect was mathematically deduced from the principles of thermodynamics a year later by Lippmann [44], and it states that a piezoelectric material will deform if an electric field is applied to it [43]. These two effects coexist in a piezoelectric material, therefore ignoring the presence of one effect in an application would be thermodynamically inconsistent [45].

The electrical behavior of piezoelectric materials can be described using Hooke’s law [46]. The electrical behavior of a material is represented by Equation (5):(5)D=ε E,
where D is the displacement of charge density, ε is the permittivity, and E is the applied electric field strength.

To define a system, Hooke’s Law states that:(6)S=s T,
where S is the strain, s is the compliance, and T is the stress.

Equations (5) and (6) are combined to form a new relationship:(7)$$\{\begin{array}{c} \left\{S\right\}=\left[s^{E}\right]\;\left\{T\right\}+\left[d\right]\;\left\{E\right\}\;\;\\ \left\{D\right\}=\left[d^{t}\right]\;\left\{T\right\}+\left[\varepsilon^{T}\right]\;\left\{E\right\} \end{array}$$
where [d] is the direct piezoelectric effect matrix, $\left[d^{t}\right]$ is the matrix which describes the converse piezoelectric effect, E indicates that a zero electric field, or a constant electric field, is found in the system, T indicates a zero stress field, or a constant stress field across the system and t determines the transposition matrix.

The four piezoelectric coefficients, $d_{ij}$, $e_{ij}$, $g_{ij}$, $h_{ij}$ are defined, as shown in Equation (8):(8)$$\{\begin{array}{c} d_{ij}={\left(\frac{\partial D_{i}}{\partial T_{j}}\right)}^{E}={\left(\frac{\partial S_{j}}{\partial E_{i}}\right)}^{T}\\ e_{ij}={\left(\frac{\partial D_{i}}{\partial S_{j}}\right)}^{E}={\left(\frac{\partial T_{j}}{\partial E_{i}}\right)}^{S}\\ g_{ij}={\left(\frac{\partial E_{i}}{\partial T_{j}}\right)}^{D}={\left(\frac{\partial S_{j}}{\partial D_{i}}\right)}^{T}\\ h_{ij}={\left(\frac{\partial E_{i}}{\partial S_{j}}\right)}^{D}={\left(\frac{\partial T_{j}}{\partial D_{i}}\right)}^{S} \end{array}$$
where the first terms are related to the direct piezoelectric effect, and the second terms are related to the converse piezoelectric effect.

A simplified method to describe both the direct and converse piezoelectric effects is represented by Equation (9) [47]:(9)$$\{\begin{array}{c} D=d\;T+\varepsilon \;E\\ S=s\;T+d\;E \end{array}$$
where S is the strain, T is the stress, E is the electric field intensity, D is the electric displacement, d is the piezoelectric coefficient, ε is the permittivity, and s is the elastic compliance.

The performance of the piezoelectric energy harvesters is dependent on the transducer’s mechanical-electrical conversion efficiency $E_{\%}$ [1]. The energy conversion efficiency for a piezoelectric transducer can be calculated using Equation (10) [48]:(10)$$E_{\%}=\frac{P_{out}}{P_{in}}\;100$$
where $P_{out}$ is the electrical output power, defined in Equation (11), and $P_{in}$ is the mechanical input power (Equation (12)).
(11)$$P_{out}=v_{p}\;i_{p}$$
(12)$$P_{in}=F\;v$$
where $v_{p}$ is the overall voltage between the transducer’s electrodes, $i_{p}$ is the current flowing through the piezoelectric transducer when the circuit is closed, F is the external mechanical force, and v is the speed of the moving object.

Another important phenomenon that must be taken into consideration when working with piezoelectric materials is the change of polarization under mechanical stress. Three factors can influence the direction and strength of the polarization: the orientation of polarization within the crystal, the crystal symmetry, and the stress applied by the mechanical deformation of the system. Any change in the polarization can be measured as the change in surface charge density at the crystal faces, and is measured in Cm^−2^ or, more commonly seen, µC/cm^2^.

A piezoelectric energy harvester has two basic parts [49]:
The mechanical module—generates electrical energy;The electrical module—comprises an electrical circuit which converts and rectifies the generated voltage.

Therefore, the effectiveness of the energy harvester is not only dependent on the piezoelectric transducer, but also on its integration with the electrical circuit [49].

Piezoelectric energy harvesting systems are in general associated with three phases (Figure 3) [1,50]:Mechanical—mechanical energy conversion: associated with the mechanical strength of the piezoelectric energy harvester under enormous stresses and the matching of mechanical impedance;Mechanical—electrical energy conversion: involves the electromechanical coupling factor of the piezoelectric energy harvester structure and piezoelectric coefficients;Electrical—electrical energy conversion: comprising electrical impedance matching

Piezoelectric energy harvesting presents a multitude of advantages:High energy and power density [5,49,51,52];Simple structure [5,49];It does not need an external voltage source [5,52];Piezoelectric materials can be meshed into hybrid materials to produce a broad range of voltages [5];Good scalability [51];Piezoelectric transducers have versatile shapes [51];Piezoelectric transducers can be easily incorporated in energy harvesting structures [53];Many piezoelectric materials have a high Curie temperature (the temperature above which the materials lose piezoelectricity) [53];Ease of application [52];It can be fabricated at both macro- and micro-scales [52].

Despite all of these advantages, piezoelectric energy harvesting systems also have a few disadvantages:The power harvested is low [5] compared with other harvesting techniques (e.g., Thermoelectric Generator - TEG devices generate up to 125 W);The harvesters require rectification, maximum power extraction, and output voltage regulation [51];Due to their high generated voltage and low output current, piezoelectric energy harvesters are not always suitable to be used with low voltage CMOS process [54,55,56].

In 1770, a French scientist, Abraham-Louis Perrelet, used the energy harvested from arm movements to design a completely autonomous, self-winding pedometer watch. This was the first record in history when somebody harvested the energy generated by body movement. Centuries later, human-based kinetic energy generators remain unexploited [5]. A healthy person takes around 10,000 steps a day [15], therefore, human walking can generate significant power if harvested [57]. Human steps have a frequency between 1.2 Hz to 2.2 Hz when considering 1.4 m/s as an average walking speed [58]. If we consider that the human walking frequency is 1 Hz, power density for piezoelectric transduction can be as high as 343 mW/cm^3^ (theoretical value) and 19 mW/cm^3^ (practical value) [5]. Therefore, in addition to providing a space for walking, the pavement can serve as a solution to recover and harvest the energy generated by walking [5].

The main challenge in developing a piezoelectric energy harvesting pavement is the low frequency of walking compared with the high operating bandwidth of piezoelectric energy harvesting systems. To obtain the best results, the operating excitation frequency must be within the range of the resonant frequency of the harvester [16]. For this, a frequency up-conversion technique must be used [29].

In addition to the frequency up-conversion, this kind of energy harvesting also needs to extract the maximum power from the piezoelectric transducer to maximize the net energy flowing into the storage device [51]. Since a piezoelectric transducer has a relatively large capacitive term with a low resonant frequency, extracting the maximum power requires an impractically large inductor (tens of hundreds Henry) [51]. A solution to this problem can be resistive matching [59,60] or using a nonlinear technique called Synchronized Harvesting on Inductor (SSHI) [61].

## 3. Piezoelectric Materials

Figure 4 [41] presents the piezoelectric materials in the crystal classes.

Non-centrosymmetric materials are materials lacking a center of inversion. There are 32 crystal classes of which 20 possess direct piezoelectricity, and 10 of these are polar crystals (in the absence of mechanical stress, they exhibit spontaneous polarization). These polar crystals will show pyroelectricity—in the presence of an oscillating thermal gradient, they will generate a charge. Moreover, the materials are ferroelectric if the dipole moment is reversible when a sufficiently large electric field is applied. Therefore, ferroelectric materials are also piezoelectric, but they exhibit semiconductor properties that are similar to the properties found in mechanically stressed piezoelectric materials [41].

There are around 200 piezoelectric materials used in energy harvesting applications [62], found in four main categories:Single crystals (Rochelle salt, lithium niobite, quartz crystals);Ceramics (barium titanate (BaTiO_3_), lead-zirconate-titanate (PZT), potassium niobate (KNbO_3_));Polymers (polylactic acid (PLA), polyvinylidene fluoride (PVDF), co-polymers, cellulose and derivatives);Polymer composites or nanocomposites (polyvinylidene fluoride-zinc oxide (PVDF-ZnO), cellulose BaTiO_3_, polyimides-PZT) [63].

Another classification of piezoelectric materials is [64]:Naturally occurring: Quartz, Rochelle salt, Topaz, Tourmaline group;Synthetic: Barium titanate, lead titanate, lithium niobite, lead zirconate titanate.

The choice of a piezoelectric material depends not only on their piezoelectric properties and the functionality of the application sector but also on parameters such as design flexibility, application frequency, and available volume [63].

Although quartz is more precise and has a high acoustic quality, it costs more than piezoceramic zirconate titanate ceramics and has a lower sensitivity, which limits the resolution of quartz charge mode sensors [65]. The most commonly used materials for piezoelectric energy harvesting devices used to be lead-based materials such as PZT. However, due to legislative measures regarding the toxicity of lead [66], PZT was replaced by other lead-free materials such as BaTiO_3_, which has a lower transduction efficiency [9]. Despite their better piezoelectric properties, some of the disadvantages associated with piezoelectric ceramics are: rigidity, brittleness, high density, lower voltage coefficient, physical limitation in producing small-sized piezoelectric ceramics, and lack of design flexibility [53,67,68].

Piezoelectric polymers are a better candidate for piezoelectric energy harvesting applications since they are mechanically flexible, so they can withstand high strain. They also generate suitable voltage with sufficient power output, despite their low power density, and they can resist higher driving fields because they have a higher dielectric breakdown and possess maximum functional field strength; they have a low fabrication cost, and the processing is faster compared with ceramic-based composites [9,53,63].

Maamet et al. [9] presented the main characteristics of piezoelectric material types, as shown in Table 1 [63,66,67,68,69,70,71,72,73,74,75,76,77,78,79,80,81,82].

The work of Mishra et al. [63] highlighted the most important parameters of piezoelectric materials, based on [83,84,85], and a brief synthesis is reproduced in Table 2.

Erturk and Inman [86] emphasized the difference between several ceramics (PZT-5A and PZT-5H) and single crystals (PMN-PT and PMN-PZT) in Table 3.

In Table 3, $d_{31}$ is the piezoelectric constant, $s_{11}^{E}$ is the elastic compliance at constant electric field, $\varepsilon_{33}^{T}/\varepsilon_{0}$ is the dielectric constant, where $\varepsilon_{0}$ is the permittivity of free space, and ρ is the mass density.

In Table 4, different power levels are presented for several piezoelectric materials, considering the transducer’s volume and application frequency.

Erturk and Inman [92] also classified piezoelectric materials as hard single crystals/ceramics and soft single crystals/ceramics. Hard single crystals/ceramics have the mechanical damping with one order of magnitude less than their counterparts, but they produce power with one order of magnitude higher than soft single crystals/ceramics for excitations at their respective resonance frequencies.

Despite its toxicity and its low conversion efficiency, lead zirconate titanate (PZT) is the most popular piezoelectric ceramic material [42]. PZT was developed in the 1950s at the Tokyo Institute of Technology, and its versions PZT-5A and PZT-5H are the most widely implemented piezoelectric ceramic materials [93]. The popularity of PZT is due to the fact that it is one of the most efficient and cost-acceptable materials [15]. To improve conversion efficiency, two mechanisms of force amplification frames are offered: concave shape [94] and convex shape [95]. For further improvement of the conversion efficiency, Wang et al. [96] used a multilayer piezoelectric stack with a flexure-free convex force amplification frame.

## 4. Types of Transducers

Piezoelectric transducers can be found in different shapes:Cantilever beam (Figure 5);Circular diaphragm (Figure 6);Cymbal type (Figure 7);Stack type (Figure 8).

The cantilever beam structure consists of a thin piezoelectric layer (or two layers) and a non-piezoelectric layer (usually a conductive metallic layer) fixed at one end to achieve a structure operating in its flexural mode (as shown in Figure 5), and is the most widely used due to its simple geometry and generation of the maximum amount of strain. If only one piezoelectric layer is bonded to the metallic layer, the configuration is called “unimorph”; if there are two piezoelectric layers, the configuration is called “bimorph”. The bimorph structure is more popular in piezoelectric energy harvesting devices because it doubles the electrical energy output, without making any remarkable changes in the device volume [63].

The circular diaphragm structure consists of a thin disk-shaped piezoelectric layer attached to a metal shim fixed on the edges of the clamping ring, as shown in Figure 6. At the core of the diaphragm is attached a proof mass to intensify the performance under low-frequency operation and to improve the power output [63].

Cymbal transducers consist of a piezoelectric layer placed between two metal end caps on both sides (Figure 7), and they are useful in applications with higher impact forces. Applying axial stress on the metal end caps, the stress is amplified and converted into radial stress, which leads to a higher piezoelectric coefficient and, therefore, higher charge generation from the piezoelectric energy harvester [97].

Stack piezoelectric transducers consist of multiple piezoelectric layers stacked over each other in a way that the poling direction of the layers aligns with the applied force (as shown in Figure 8), and are used in applications which demand high pressure [63].

The advantages and disadvantages of different types of piezoelectric transducers are summarized in Table 5 [63].

For vibration energy harvesting, the most studied piezoelectric structures are unimorph and bimorph cantilever configurations [53]. For example, Rundy et al. [14] used a 1.75 cm PZT bimorph cantilever to harvest energy from low-level vibrations. The harvester’s natural frequency was 100 Hz, and they attached a proof mass to the tip of the cantilever to lower the resonance frequency. In this way, they obtained 60 µW of power. In another example, Sodano et al. [98] used a 63.5 × 60.3 × 0.27 mm PZT-5H cantilever on an electromagnetic shaker at 50 Hz charging a 1000 mAh NiMH rechargeable battery to 90% of its capacity within 22 h (or 160 mW average power).

Jasim et al. [99] used another type of transducer to harvest the energy produced by trucks driving on the urban expressway Route 55 in New Jersey. In a previous paper [100], they optimized the geometry considering the balance of energy harvesting and mechanical stress, resulting in a bridge transducer. The bridge transducer consists of a square PZT5X disk placed between two metal end caps and was designed to have a layered poling pattern and electrode configuration that increased the piezoelectric coefficient and capacitance, producing 2.1 mW under repeated loading of 270 kg.

## 5. Piezoelectric Modeling

For piezoelectric energy harvesting modeling, it is essential that the model represents both direct and converse piezoelectric effects. Therefore, it must show both forward and feedback interaction between the electrical and mechanical domains [48]. In addition, the model should not only represent the piezoelectric transducer, but it also should be integrated with the electrical circuit [49]. Hence, for design optimization and comprehension of the behavior, it is important to use an electromechanical model [101].

### 5.1. Piezoelectric Working Modes

When a piezoelectric energy harvester is modeled, one of the characteristics that must be represented is the working mode. There are three piezoelectric working modes [102,103]:
Transverse mode ($d_{33}$ mode);Longitudinal mode ($d_{31}$ mode);Piezotronic mode.

In general, piezoelectric transducers consist of several layers of piezoelectric, elastic, conducting, or insulating materials. When an electric potential is applied between the conducting layers, an actuation force is produced in the piezoelectric layers, and the direction of stress-strain components and the electric field indicate the modes of operation (transverse or longitudinal), as shown in Figure 9 [16].

In Figure 9a, the standard form of a piezoelectric transducer working in transverse mode is shown. An example of a piezoelectric transducer working in d_33_ mode is the stack type transducer. The working mode is based on the fact that the transducer has several piezoelectric thin films connected in parallel, and the mechanical connection is serial, layered upon each other. Because of the thin films mechanically connected in series, the total displacement of the piezoelectric stack transducer is the product between the total number of films and the movement of each film [16].

Longitudinal mode (presented in Figure 9b) was widely studied using cymbal, unimorph, and bimorph transducers. In this mode, piezoelectric thin films are fixed upon a supporting structure and placed between the electrode layers. When an electric field is applied in a vertical direction, a stress/strain is produced along the horizontal direction.

The last working mode, piezotronic, appeared with the discovery of ZnO nano wires with *n*-type conductivity [104]. In piezotronic working mode, due to the piezoelectric potential, a Schottky barrier is created between the piezoelectric nanowire and the electrode that regulates the flow of electrons [103].

### 5.2. Input–Output Dependency

A piezoelectric energy harvester can be modeled like a black box, where the dependency between the input and output characteristics is known. For example, the piezoelectric transducer’s thickness is closely related to the generated output voltage [105], therefore a piezoelectric energy harvester model should represent this dependency and highlight the critical thickness where the output voltage becomes extremely small [68]. Such a dependency is presented in [106], where the authors defined the open circuit output voltage V of a piezoceramic based energy harvester. Because the environment’s frequency range is less than 100 Hz, which is outside the frequency range of a piezoceramic transducer [98,107], they defined the output voltage in the off-resonance condition as [106]:(13)$$V=E\;t=-\frac{g\;F\;t}{A}$$
where E is the electric field, t is the piezoceramic’s thickness, A is the piezoceramic’s area, and g is the piezoelectric voltage coefficient, expressed as in Equation (14):(14)$$g=\frac{d}{\varepsilon_{0}\varepsilon_{r}}$$
where d is the piezoelectric charge constant, $\varepsilon_{r}$ is the relative dielectric constant and $\varepsilon_{0}$ is the vacuum dielectric constant.

According to Equations (13) and (14), g reflects the sensitive capabilities of piezoelectric materials [108], therefore a material with a higher piezoelectric voltage coefficient is expected to generate a higher voltage.

If the input force is a bending stress, the open circuit output voltage $V_{3j}$ is defined as presented in Equation (15) [109]:(15)$$V_{3j}=\sigma_{xx}\;g_{3j}\;L_{j}$$
where $\sigma_{xx}$ is the bending constant, $g_{3j}$ the piezoelectric constant and $L_{j}$ is the distance between the electrodes. Therefore, depending on the type of traductor, it represents either the thickness of the piezoelectric film or the lateral gap between the electrodes. Similar to Equation (14), $g_{3j}$ is defined as presented in Equation (16):(16)$$g_{33}=\frac{d_{33}}{\varepsilon_{0}\varepsilon_{r}}$$

Another critical parameter that has to be defined in a piezoelectric transducer’s model is the short-circuit current I, which can be defined using the time derivative of the displacement field D, as presented in Equation (17) [110]:(17)$$I=\frac{dD}{dt}=d\frac{dT}{dt}=d\frac{dS}{dt}$$
where T is stress, and S is strain.

Based on Equation (17), the short-circuit current depends on the rate of stress application and the resulting strain rate of the piezoelectric material. Therefore, a higher current can be obtained by applying stress more rapidly.

So far, we have formulas for the open circuit voltage and for the short-circuit current. Both are useful pieces of information for a piezoelectric energy harvester model, but since these two values are measured in completely different scenarios, the output power cannot be defined as their product. The most basic way to determine the maximum power output P of an energy harvester is to measure the voltage V across a range of resistive loads connected to the device and the current I through them [111]:(18)$$P=\frac{V^{2}}{R}=I^{2}\;R$$

In Equation (18) the maximum output power is defined. If the average output power is the one needed for the model representation, it has to be calculated depending on the type of the input force. For example, if a regularly oscillating AC source is applied at the input of the harvester, the average power is calculated using Equation (19) [111]:(19)$$P_{avg}=V_{RMS}\;I_{RMS}=\frac{V_{peak}}{\sqrt{2}}\;\frac{I_{peak}}{\sqrt{2}}=\frac{V_{peak}\;I_{peak}}{2}$$

Therefore, in this case, the RMS power is half of the peak power. Nevertheless, if the input force is an impulse-type excitation, the average power should be calculated using Equation (20) [111]:(20)$$P_{avg}=\frac{1}{T_{pulse}}\int_{t_{1}}^{t_{2}}\frac{V{\left(t\right)}^{2}}{R}dt=\frac{1}{T_{pulse}}\int_{t_{1}}^{t_{2}}I{\left(t\right)}^{2}\;R\;dt$$
where R is the resistive load connected on the device, $t_{1}$ and $t_{2}$ are the start and the end moments of the pulse, respectively, and $T_{pulse}$ is the pulse duration.

When the input force is an impulse-type excitation, the duration time of the generated output voltage is too short to collect energy, therefore, a capacitor placed at the harvester’s output is recommended. In this case, the generated energy E can be calculated using the output voltage V and the capacitance C, as shown in Equation (21) [68]:(21)$$E=\frac{1}{2}\;C\;V^{2}$$

### 5.3. Impedance Analysis

Another important electrical parameter that has to be represented in a piezoelectric transducer model is the impedance. An impedance analysis consists in representing the resistive impedance (frequency-independent component, $Z_{Re}$) and reactive impedance (frequency-dependent component, $Z_{Im}$) using a Nyquist plot. In this way, the impedance properties of the harvester can be visualized. Usually, a piezoelectric energy harvester is modelled as an RC circuit, which gives a semi-circular relationship between $Z_{Re}$ and $Z_{Im}$, represented in Figure 10 [112], where the real (resistive) impedance is equal to the diameter of the curve on the real axis and the value of capacitance can be found from the frequency $f_{C}$, where $-Z_{Im}$ is at a maximum.

The impedance analysis is used to identify the optimum load impedance for maximum power transfer, as this will be close to the internal impedance of the harvester. By modeling the piezoelectric energy harvester as an RC circuit, the resistance and capacitance are obtained from the Nyquist plot, using Equation (22):(22)$$R\;C=\frac{1}{2\pi f_{c}}$$

### 5.4. Lumped Parameter Electromechanical Model

Most models follow a cascade model structure, where decoupled sub-models are connected to describe the behavior of a piezoelectric transducer. These cascade models take into consideration only one-way coupling since they are control-oriented approximations [113], but a model should represent the energy harvesting and also the damping control of the piezoelectric device. Therefore, a physical electromechanical interpretation should be compatible with a two-port network model [114] and capable of incorporating two-way piezoelectric coupling effects [48]. Goldfarb and Celanovic [115,116] proposed in their work the lumped parameter electromechanical model, managing to postulate the nonintuitive behavior of piezoelectric transducers. An extended lumped parameter was proposed by Ruderman et al. [117], where they incorporated state-varying capacitance and Voigt–Kelvin-type linear creep effects into the model.

If we consider the piezoelectric transducer as a one-mass m with stiffness k and damping b, the mechanical domain is governed by the differential equation (Equation (23)) [48]:(23)$$m\;\ddot{x}+b\;\dot{x}+k\;x=F_{t}+F\;$$
where $F_{t}$ is the transduced force from electrical domain and F is the external mechanical force.

Piezoelectric materials are dielectric; therefore, the electric domain is mainly defined by a capacitive behavior. As was experimentally observed, between voltage and displacement, as well as between force and displacement, is exhibited a rate-independent hysteresis. This hysteresis is observed in closed leads between displacement and charge, allowing the postulation that the net electrical charge $q_{p}$ across the piezoelectric transducer can be defined as the sum between two components, as shown in Equation (24) [48]:(24)$$q_{p}=q_{c}+q_{t}$$
where $q_{c}$ is defined in (25) and $q_{t}$ is defined in Equation (26).
(25)$$q_{c}=C\;v_{t}$$
where C is the electrical capacitance of the transducer and $v_{t}$ is the voltage across the capacitor.

Because of the direct piezoelectric effect, there is a coupled charge induced by the mechanical domain due to the relative displacement x. Furthermore, because of the converse piezoelectric effect, there is a transduced force from the electrical domain proportional to the capacitor voltage, defined in Equation (27). Considering the constant ratio T, the electromechanical coupling between mechanical and electrical domain is given by [48]:(26)$$q_{t}=T\;x$$
(27)$$F_{t}=T\;v_{t}$$

Rate-independent hysteresis exists only in the electrical domain, between the applied actuator voltage and resulting charge, thereby introducing non-linearities into the electrical domain, as shown in Equation (28):(28)$$v_{h}=H\;\left(q\right)$$

The overall voltage between the transducer’s electrodes is given by Kirchhoff’s second law, highlighted in Equation (29):(29)$$v_{p}=v_{t}+v_{h}$$

From the lumped parameter electromechanical model, represented in Equations (23)–(29), it is possible to draw the piezoelectric transducer that couples the electrical and mechanical domains, as shown in Figure 11, where $i_{p}$ = ${\dot{q}}_{p}$ is the current flowing through the transducer when the circuit is closed.

### 5.5. The Finite Element Method

In literature, several approaches to develop piezoelectric energy harvester models are presented, but two of them are the most widely used, showing a close correlation between theoretical and experimental data [118]:The analytical distributed parameter model;The finite element model.

The first was introduced by Erturk and Inman [119] and is applied for unimorph and bimorph piezoelectric energy harvesters to obtain a modal solution of second-order ordinary differential equations. The second was adapted by De Marqui Junior et al. [120,121], and uses standard discretization to provide discrete models with less constraining hypotheses on the global electrical variables.

Aloui et al. [118] studied the finite element modeling method of a piezoelectric composite beam with *P* piezoelectric layers and used a standard discretization of *N* mechanical degrees of freedom. They expressed the general damped electromechanical equations, as shown in Equations (30) and (31), where Equation (30) corresponds to the mechanical equation of motion and Equation (31) corresponds to the electrical equation.
(30)$$M_{m}\;\ddot{U}\left(t\right)+c_{m}\;\dot{U}\left(t\right)+K_{m}\;U\left(t\right)+K_{c}\;V\left(t\right)=F\left(t\right)$$
(31)$$K_{e}\;V\left(t\right)-K_{c}^{T}\;U\left(t\right)+Q\left(t\right)=0$$
where U(t) is the (*N* × 1) vector of the mechanical coordinate displacements, V(t) is the (*P* × 1) voltage vector, Q(t) is the (*P* × 1) electric charge vector, $M_{m}$ is the (*N* × *N*) mass matrix, $c_{m}$ is the (*N* × *N*) damping matrix, $K_{m}$ is the (*N* × *N*) stiffness matrix, $K_{c}$ is (*N* × *P*) electromechanical coupling matrix and $K_{e}$ is the diagonal (*P* × *P*) capacitance matrix.

Using Equation (30), the natural frequencies of the piezoelectric energy harvester can be calculated. Piezoelectric energy harvesters have two natural frequencies, which are defined as characteristic limit bounds for each vibration mode:The short-circuit natural frequencies—where it is assumed that no potential difference exists across the piezoelectric layers in free vibration, therefore the electromechanical coupling in Equation (30) is omitted [118];The open circuit natural frequencies—where it is assumed that no charge flows in the electrical circuit [122].

In practice, the short-circuit condition corresponds to a low resistive load connected to the electrodes of the piezoelectric layers (R→0), and the open-circuit condition corresponds to a high resistive load (R→∞).

For the finite element method, the sensitivity analysis is a useful tool for modeling and designing analysis of mechatronic systems and is applied to solve engineering problems such as structural analysis, model updating, design optimization of structures, system control, and uncertainty propagation [123]. The method enables the evaluation of the degree of influence of the input parameters on the output responses of piezoelectric models, and is considered the most critical step in design optimization [124].

Sensitivity analysis techniques are classified into:The local method, also referred to as one factor at time analysis—based on the approximation of partial derivatives to evaluate how uncertainty in one factor affects the response of the model keeping the other factors fixed at their nominal values;The global method—evaluates the effect of parameters that are varying within the considered multidimensional space [125], as seen in the Aloui et al. [118] work mentioned above.

### 5.6. Equivalent Electric Circuit Model

Rjafallah et al. [1] modeled a PU-50%-vol%-PZT composite using the equivalent electric scheme presented in Figure 12.

In the equivalent circuit, there are two blocks interconnected by $R_{i}$, which represents the electrical connection between PU and PZT-ceramic materials. One block models the behavior generated by vibrations, where $I_{ac}$ is the generated current, $C_{e}\left(\omega \right)$ is the capacitance of PU and $R_{e}\left(\omega \right)$ is the resistance representing the dielectric losses of the PU matrix, and the other block models the short-circuit current $I_{cc}$, the capacitance $C_{p}\left(\omega \right)$, and the dielectric losses of PZT-ceramic particles $R_{p}\left(\omega \right)$. $V_{dc}$ is the bias voltage source, $R_{s}$ is the resistance of the electrodes and wires, L is the inductance of the liaisons, and R represents the electrical resistance of the electrical load.

The total current $I_{T}$ traversing the resistance R is obtained using the superposition theorem, presented in Equation (32):(32)$$I_{R}^{\left(T\right)}=I_{R}^{\left(1\right)}+I_{R}^{\left(2\right)}+I_{R}^{\left(3\right)}$$
where $I_{R}^{\left(1\right)}$ is the static electric current provided with the voltage source $V_{dc}$ when all current sources from Figure 12 are replaced by open circuits, $I_{R}^{\left(2\right)}$ is the current provided during the electrical-mechanical conversion, performed by PU, where $I_{ac}$ is active while $I_{cc}$ is replaced by open circuit and $V_{dc}$ is replaced by short-circuit, and $I_{R}^{\left(3\right)}$ corresponds to the electric current due to the mechanical-electrical conversion, carried out by the PZT-ceramic particles, when $I_{cc}$ is active while $I_{ac}$ is replaced by open circuit and $V_{dc}$ is replaced by short-circuit. $I_{R}^{\left(2\right)}$ and $I_{R}^{\left(3\right)}$ have the same frequency as that of the mechanical vibration. Taking into consideration all of these, Equation (32) is equivalent to Equation (33) [1]:(33)$$I_{R}^{\left(T\right)}=\frac{V_{dc}}{R_{s}+R_{e}+R_{i}+R_{p}+R}+\frac{Z_{e}\;I_{ac}}{R_{s}+jL\omega +Z_{e}+R_{i}+Z_{p}+R}+\frac{Z_{p}\;I_{cc}}{R_{s}+jL\omega +Z_{e}+R_{i}+Z_{p}+R}\;$$

The root mean square harvested power can be calculated using Equation (34) [1]:(34)$$\;P_{RMS}=\sqrt{\frac{1}{T}\int_{0}^{T}{\left(R\;i^{2}\left(t\right)\right)}^{2}dt}=\sqrt{\frac{1}{T}\int_{0}^{T}{\left(R\;{\left(i_{R}^{\left(2\right)}\left(t\right)+i_{r}^{\left(3\right)}\left(t\right)\right)}^{2}\right)}^{2}dt}$$
where $i_{R}^{\left(2\right)}\left(t\right)$ and $i_{R}^{\left(3\right)}\left(t\right)$ are the temporal electric currents corresponding to the complex electric currents $I_{R}^{\left(2\right)}$ and $I_{R}^{\left(3\right)}$.

The model can be simplified by removing $R_{s}$ and L from the equivalent electric scheme due to their weak values compared to the electrical impedances of PU and PZT [1].

## 6. Frequency Response

One of the main characteristics in piezoelectric energy harvesting is the frequency response, since the energy harvesters perform best when their resonance frequency matches their input frequency [63,126]. Currently, most piezoelectric energy harvesters are resonance-based devices; this means that the harvester’s resonant frequency matches the source’s frequency in order to achieve high efficiency [127]. Otherwise, a small mismatch can generate a significant reduction in voltage and power output [63]. Therefore, the size and shape of the piezoelectric layers are designed according to the natural frequency of the system and the piezoelectric material is chosen to match the application frequency. For example, piezoelectric ceramics are used in applications that demand high vibration frequencies (above 100 Hz) due to their high stiffness and superior piezoelectric properties. In comparison, piezoelectric polymers and composites are used in applications that require lower vibration frequencies (up to 30 Hz) [63].

There are two types of frequency response functions [12]:The frequency response function of linear field variables (FRF) or mobility function: is a function of the excitation force and output response velocity;The frequency response function for power variables (FRFP): is a function of output power or squared voltage and input power.

In order to maximize the harvested power, the device’s resonant frequency should correspond to the ambient frequency, but the energy of ambient vibrations is often distributed over a wide frequency spectrum. Therefore, several optimization techniques are used to broaden the harvester bandwidth [9]. Usually, PZT sensors are characterized by a large bandwidth, e.g., Hemmasian Ettefagh et al. [128] studied a sensor that showed high performance within the 32–6400 Hz sensing bandwidth.

To adjust the piezoelectric energy harvester’s resonance frequency after fabrication, natural frequency tuning can be implemented using different techniques [9]:Geometrical tuning—the technique consists of adjusting some of the system’s geometrical parameters (inertial mass [129,130] and shape [131,132]) to tune its resonance frequency. The technique implies a simple design without affecting the damping, and before installation, a fine-tuning is possible.Preload application—the stiffness is reduced by applying an axial preload to the beam. The technique has a larger effective operating region, but it affects the damping, and it is not suitable for fine-tuning [9,133].Extensional mode resonator—the concept uses extension deformation and adjusts the distance between the vibrating beams. The technique has a large effective operating region but also implies complex design, and it is delicate for fine-tuning [9,134].Stiffness variation—a magnetic stiffness is added to the system. The technique is easy to implement and has a simple design, but by adding a magnet, the resonance power is reduced, and it also has a complicated nonlinear behavior [9,135,136,137,138].

To increase the operating frequency range, a multi-frequency system, which consists of an array of sub-systems with their own resonance frequency, is needed. There are two techniques for a multi-frequency sub-system [9]:Cantilever array—the technique uses an array of cantilevers with different dimensions, each with its own resonance frequency. The concept allows having better control over the final frequency range by adjusting the number and dimensions of the cantilevers, it has a simple design and a uniform frequency spectrum, but the total volume is increased [9,139].Multimodal system—several seismic masses are used in order to have different vibration modes with different resonance frequencies. The technique implies a complex design and a non-uniform frequency spectrum, while the total volume is increased [9,140].

Another optimization method is to use a nonlinear system that can be implemented using [9]:Levitation based systems—the technique uses magnetic repulsion to keep a permanent magnet in levitated oscillations. It is used for low-frequency energy harvesting, and it has no mechanical friction, but the concept is rarely used in piezoelectric energy harvesting [9,141,142,143].Vibrating beam with magnetic interaction—the technique combines vibrating beams with magnetic interactions by exploiting nonlinear magnetic forces to change the dynamics of a vibrating structure. It can adjust the frequency response, but the design is complex, and the total volume is increased [9,144,145,146].Mechanical nonlinearity—the nonlinearity is created by a preloaded clamped-clamped structure. The technique is used for frequency tuning with preload configuration, and it does not need an external system, but the design is complex [9,147,148,149,150].Amplitude limiter configuration—the technique uses a mechanical stopper to broaden the operating frequency range by changing the damping and the stiffness of the system at impact. The design is simple, but it has energy losses due to mechanical impact and risks of mechanical impact fatigue [9,151,152,153].

All of the widen operating frequency systems presented above are resonant, and most operate at frequencies above 50 Hz. Below 50 Hz, energy harvesting with resonant systems becomes more challenging. Moreover, below 10 Hz, using a resonant system becomes unrealistic. Therefore, for this frequency range, non-resonant systems are used, considering two approaches: frequency up-conversion and free moving mass [154].

Frequency up-conversion systems combine a low-frequency system (which absorbs the energy when a low-frequency excitation occurs) and a high-frequency system (which absorbs the energy transferred by the low-frequency system and converts it into electrical energy). However, frequency up-conversion systems need a relatively high amplitude of excitation in order to have an interaction between the low-frequency and high-frequency systems. For this, the following techniques are used [9]:Resonators-based frequency up-conversion—both low-frequency and high-frequency systems are resonant, and most of them consist of vibrating beams with high and low inertial mass, respectively. The design is simple, and it has a wide bandwidth, but the total volume is increased, and it has risks of mechanical impact fatigue.Free moving ball-based frequency up-convertors—in this technique, the low-frequency system is replaced by a free moving ball which transfers energy to a vibrating structure. The design is simple, and it harvests energy from arbitrary motion, but it has a non-uniform frequency spectrum, and it is rarely used in piezoelectric energy harvesting.

Unlike the frequency up-conversion systems, which require at least one resonating structure for transduction, free moving mass harvesters are entirely non-resonant. Their principle is based on the arbitrary motion of a mass and is used for harvesting from very low frequencies (below 10 Hz), such as human motion. There are two free moving mass techniques:Free moving object—the harvester has a free moving object that can be suspended by a rope [155], a rod [156], or rolling inside a cage [157]. The design is simple and generates relatively high power, but the ball’s movements are unpredictable.Free moving liquid—the technique uses ferrofluid motions to vary a magnetic field across a coil. The design is simple and it detects infinitely low displacement, but it generates low power and the technique is rarely used in piezoelectric energy harvesting.

Since the mass movement is arbitrary and multidirectional, these techniques also imply 3D harvesting. Three-dimensional harvesting systems harvest energy from all directions (translation and rotational movements) but, unfortunately, this technological area is the least found in the literature [9].

## 7. Piezoelectric SPICE Models

As presented in the previous sections, piezoelectric transducers are classified into numerous types, use different materials, and their behavior is dependable on the input parameters. Therefore, choosing a suitable transducer for different applications implies the use of several SPICE models.

Mouapi and Hakem [158] developed a piezoelectric SPICE model for a QP20W transducer, which consists of a sine current source with a parasitic capacitance and resistance, as presented in Figure 13.

Another approach is to use a sine voltage source to simulate a piezoelectric transducer, as Linear Technology did for the MIDE V22BL transducer [159] (Figure 14).

## 8. Electronic Circuits for Piezoelectric Energy Harvesting

As previously explained, when an external mechanical force is applied to a piezoelectric transducer, it will generate an electric field due to the direct piezoelectric effect. However, the generated energy varies due to the speed and magnitude of the input force. This means that the only application in which the piezoelectric transducers can be used alone is measuring the input force unless the input force is mechanically regulated, which implies external power. Therefore, piezoelectric energy harvesting applications do not use stand-alone transducers, they need to rectify and store the output to use the generated energy in external sensors and transmitters [160]. For this purpose, different electronic circuits can be used (the most popular techniques are summarized in Table 6), but there are also several commercially available integrated circuits for power management optimization, e.g., MB39C811, LTC3588-1, LTC3599-2, and MAX17710 [161].

The circuits presented in Table 6 are detailed in the next subsections.

### 8.1. AC-DC Piezoelectric Energy Harvesting Circuit

To rectify the AC power generated by the piezoelectric transducers, an AC-DC energy conditioning circuit is needed between the transducer and the energy storage [15]. There are several electrical schemes used to obtain the highest energy conversion efficiency, but the most popular circuit uses a bridge rectifier to convert the output voltage of both polarities to a single polarity. This circuit is used to charge a capacitor $C_{L}$ to a required voltage, after which a switch S is closed, connecting the capacitor to the load $R_{L}$, as presented in Figure 15 [15,160], where the piezoelectric element is modeled as an equivalent current source $I_{eq}$ in parallel with a capacitor $C_{p}$ and the internal leakage resistance $R_{p}$.

The AC-DC energy harvesting circuit can also be represented using a lumped-parameter electromechanical model with a single degree of freedom, as presented in Figure 16 [162,163], where u(t) is the transverse displacement of the transducer, F(t) is the excitation force, M is the effective mass, K is the effective stiffness, η is the effective damping coefficient, Θ is the effective piezoelectric coupling coefficient and $C_{p}$ is the effective capacitance.

### 8.2. Two-Stage Piezoelectric Energy Harvesting Circuit

Since the excitation force level is not constant, the rectified voltage level is not constant either. Therefore, a DC-DC converter can be used after the rectifier to maximize the power transferred to the storage device [59,164]. Another solution is to use a step-down DC-DC converter operating in discontinuous conduction mode (DCM) if the generated voltage is higher than the required voltage level of the storage device [164,165]. In this case, if the duty cycle is optimized for the step-down converter, the energy flow to the battery can be tripled [165]. In Figure 17, Guan and Liao [166] presented an optimized two-stage energy harvester, where $C_{0}$ is a temporary storage capacitor.

The DC-DC converter block is presented in detail in Figure 18 [166]:

For the circuit presented in Figure 18, the converter efficiency $\eta_{C}$ is calculated in Equation (35) [166]:(35)$$\eta_{C}=\frac{V_{rect}+V_{D}-V_{ces}}{V_{rect}}\;\frac{V_{esd}}{V_{esd}+V_{D}}$$
where $V_{rect}$ is the rectified voltage, $V_{D}$ is the forward bias of the diode $D_{1}$, $V_{ces}$ is the voltage drop of the internal switch $T_{r}$ and $V_{esd}$ is the voltage of the energy storage device.

### 8.3. Synchronized Switch Harvesting on Inductor (SSHI)

If the system is weakly coupled (and not only in this situation), the harvested power can be increased using a nonlinear technique called Synchronized Switch Harvesting on Inductor (SSHI) [167]. This technique is based on the resonant circuit created by the internal capacitor of the piezoelectric transducer and an external inductor which flips the capacitor voltage instantly to nullify the effect of the capacitive term. In this way, the energy used to charge the internal capacitor in order to conduct the diode bridge, which would have been wasted, is now harvested [51].

In the classical AC-DC energy harvesting circuit (Figure 15), a negative power is produced because the output current and generated voltage cannot keep the same phase; this means that the harvested energy may return to the mechanical part, losing some of the harvested power [168]. The SSHI circuit (Figure 19 and Figure 20) overcomes this problem by adding a switch path (which contains the switch $S_{1}$ and the inductor $L_{1}$). The switch opens instantly when the capacitor voltage reaches the maximum in the opposite polarity to result in the flipping of the capacitor voltage. This enables the circuit to harvest the capacitor charge instead of being discharged to waste [51].

There are two SSHI circuits: parallel (Figure 19) and series (Figure 20). Both can increase the output voltage [169]; the only difference between them is whether the switch path is connected in parallel or in series with the rectifier [15].

Lallart and Guyomar [169] designed a self-powered SSHI interface able to automatically perform switching actions once the output voltage reaches its maximum. The schematics of the self-powered parallel-SSHI and series-SSHI are presented in Figure 21 and Figure 22.

In Figure 21 and Figure 22, the diodes are 1N4004 model, $T_{1}$ and $T_{3}$ are TIP32C, $T_{2}$ and $T_{4}$ are TIP31C, $L_{1}$ has 120 mH and 97 Ω, $C_{1}$ and $C_{2}$ are 23 nF capacitors and $C_{L}$ has 47 µF [169].

For the self-powered series-SSHI, a current flowing through the circuit is generated by the piezoelectric transducer. When the output voltage across the piezoelectric transducer is increasing, $C_{1}$ will be charged, while $T_{1}$ and $T_{2}$ are blocked, and when this voltage reaches the peak value, it starts decreasing and, between the emitter and the base of $T_{1}$, appears a voltage difference. When this voltage difference reaches the threshold voltage of $T_{1}$, it will start conducting and at the same time, $C_{1}$ provides the voltage for $T_{2}$’s base since $D_{1}$ is blocked. The conduction of $T_{1}$ and $T_{2}$ initiates the inductance-capacitance resonant circuit and $C_{p}$ is quickly discharged through $D_{3}$, $\;T_{2}$, $R_{L}$, $D_{8}$, and $L_{1}$. With the presence of inductor $L_{1}$, the voltage across $C_{p}$ is inverted. Similarly, when the voltage reaches the negative peak, it will also be inversed due to the circuit’s symmetrical topology [15].

For the self-powered parallel-SSHI, the working principle is similar. The difference consists in the fact that when $C_{p}$ is discharging, the current flows only through $D_{3}$, $T_{2}$ and $L_{1}$, or $D_{4}$, $T_{3}$ and $L_{1}$. This means that the voltage inverting process is not influenced by $R_{L}$.

### 8.4. Synchronous Electrical Charge Extraction (SECE)

Another method for piezoelectric energy harvesting electrical circuits is Synchronous Electrical Charge Extraction (SECE). Unlike other methods, the generated power using the SECE method does not depend on the load, therefore, the load can vary without affecting its efficiency. The disadvantage of this method consists of the complex controller of the circuit [51].

The electrical circuit of the SECE method is presented in Figure 23 [170].

In Figure 23, after the AC-DC converter, which consists of four UF4004 rectifier diodes, a DC-DC power converter was implemented using a flyback switching mode converter. The latter uses an IRFD220 MOSFET transistor T, a Myrra 74,010 coupled inductor L, an SB540 Schottky diode $D_{5}$ and a 1000 µF storage capacitor $C_{L}$ [170].

The gate voltage of the MOSFET transistor, $u_{g}$, is determined by a control circuit measuring the rectified voltage $V_{rect}$, and controls the power converter operation. When the rectified voltage reaches the peak value, a 15 V voltage is applied on the T gate, the transistor starts conducting and transfers the electric charge from the piezoelectric transducer to the coupled inductor. When the piezoelectric transducer’s electrical charge is completely extracted, the control circuit detects the cancellation of the rectified voltage, and T is blocked. Now that the transistor is blocked, the piezoelectric transducer returns to an open circuit, and the energy stored by L is transferred to the storage capacitor. When the rectified voltage reaches the peak value again, synchronously with mechanical displacement, the next electric charge extraction sequence occurs [170].

In theory, with determined input power, the flyback converter delivers a constant output power independent of the load. However, in practice, due to the components’ imperfections, the effective output voltage domain is restricted [170].

## 9. Piezoelectric Energy Harvesting Applications

In the literature, numerous examples of applications for piezoelectric energy harvesting exist. For example, Liu et al. [15] used a monolithic multilayer piezoelectric stack manufactured by Smart Material, which consists of 300 layers of piezoelectric patches with the internal capacitance of 2.1 µF, to test the efficiency of a footstep piezoelectric-stack energy harvester, in a laboratory experiment. The piezoelectric device was placed in an amplification frame stimulated by a shaker. The input force was measured by a force transducer placed between the amplification frame and the shaker, while the amplified force was measured by another force transducer placed between the interface of the piezoelectric stack harvester and the amplification frame. They considered the force excitation of human locomotion 114 N, and the amplified force resulted being 846 N. With this setup, they used three different circuits:When they used a standard energy harvesting circuit (a simple AC-DC converter) with a 50 kΩ load, they obtained an output power of 1.35 mW;When they used a series-SSHI circuit with a 60 kΩ load, they obtained an output power of 1.33 mW;When they used a parallel-SSHI with a load of 80 kΩ, they obtained an output power of 2.35 mW, and the harvesting efficiency was increased by 74% in comparison with the first circuit.

A different way to achieve this result without using a shaker is to use a common loudspeaker excited by an audio power amplifier. This is a convenient way to modify the output frequency and the excitation level.

He et al. [125] designed a piezoelectric energy harvester structure using a double-layer squeezing structure and a piezoelectric beam array. During the experiment, a 60-kg person stepped on and off the structure with different frequencies. The maximum output power obtained by one piezoelectric beam was 134.2 µW under a step frequency of 1.81 Hz, but the authors considered that a total of 40 piezoelectric beams inside the floor structure could reach 5.368 mW.

Zhang et al. [3] integrated a stiffness spring with an energy generator to develop a device for harvesting the mechanical energy of the foot. The harvester was placed in a shoe near the heel, therefore, when the person’s foot touches the ground, the heel compressed a pedal, and a piezoelectric beam was bent, generating electrical energy. When the foot lifted off the ground, the stiffness spring already stored elastic potential energy, which was converted to kinetic energy, bending the piezoelectric beam once again. A 60-kg person could obtain 235.2 mJ per step.

A comparison study of vibrational energy harvesting using piezoelectric tiles in walkways vs. stairways was presented in [171]. The authors concluded that piezoelectric tiles placed in a stairway perform better than the ones placed in walkways due to the natural increased pedestrian work demanded in traversing the stairs. They also indicated that the tile design should consider the naturally random characteristics of pedestrian traffic in order to increase the level of harvested power.

Cho et al. [13] developed a road-compatible piezoelectric energy harvester using piezoelectric transducers fixed onto both ends, the stress being converged toward the center of the device via a rigid bar. The harvester was used in actual road conditions for five months, being stressed by the vehicles traveling at speeds of 10–50 km/h when they entered a highway rest area. At 50 km/h, the harvester’s output power was 2.381 W, while at 10 km/h, the output power was 576 mW. The generated energy was used to power the LED indicators and to transmit real-time information about the sensor’s leak, temperature, and strain.

Cha et al. [172] analyzed the possibility of harvesting the energy generated by the mouse click motion, using a unimorph PVDF beam structure. The maximum harvested energy, obtained when the load resistance matched the impedance of the piezoelectric transducer for the fundamental harmonic, was in the range of 1–10 nJ, but this level could be increased by using multiple piezoelectric layers or alternative materials with higher efficiency.

## 10. Conclusions

The development of the Internet of Things concept, wearables devices, and wireless technologies has led to the need for self-powered systems due to the inaccessibility of batteries for changing. A solution for these self-powered systems is to harvest mechanical energy using piezoelectricity. Piezoelectric materials have the property to generate an electric field when a mechanical force is applied. This phenomenon is known as the direct piezoelectric effect.

Piezoelectric energy harvesting has several advantages, such as high energy and power density, low cost, good scalability, and ease of application. However, due to its main disadvantages (low level of harvested power and the need for rectification, maximum power extraction, and output voltage regulation), piezoelectric transducers cannot be used alone to harvest mechanical energy. Therefore, a piezoelectric energy harvester usually contains an AC-DC converter, has a two-stage conversion circuit, or uses non-linear techniques such as SSHI or SECE.

The piezoelectric energy harvesting technique can be used in a multitude of applications; therefore, each implementation needs to optimize the technique for its own needs. First, a suitable transducer must be found. Piezoelectric transducers can be found in different shapes (cantilever beam, circular diaphragm, cymbal type, stack type, and more) and can be made from different materials, each with its own characteristics. When the piezoelectric transducer is chosen, the next step is to develop a model in order to simulate and optimize the behavior in the integrated system.

## Figures and Tables

**Figure 1 sensors-20-03512-f001:**
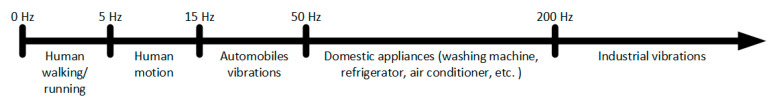
Frequency level for different mechanical energy sources [9].

**Figure 2 sensors-20-03512-f002:**
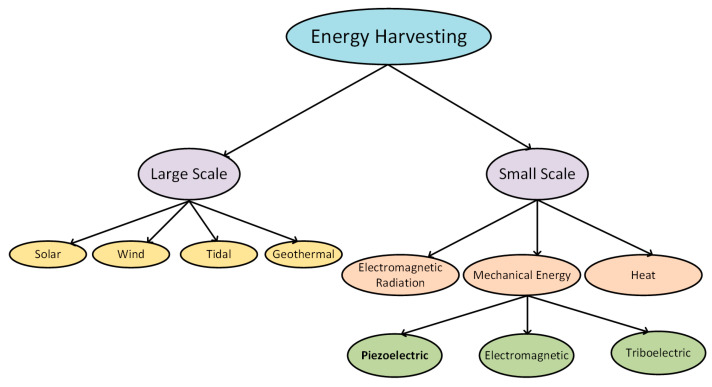
Classification of energy harvesting sources.

**Figure 3 sensors-20-03512-f003:**
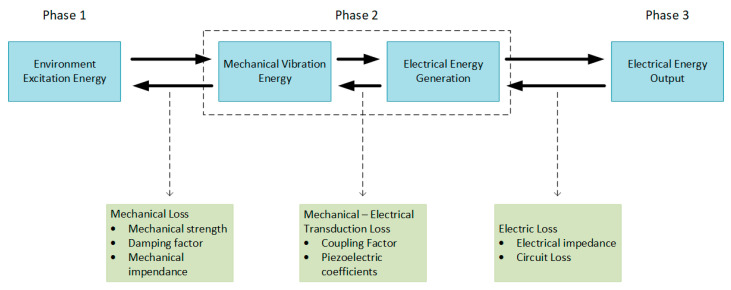
The three phases associated with piezoelectric energy harvesting.

**Figure 4 sensors-20-03512-f004:**
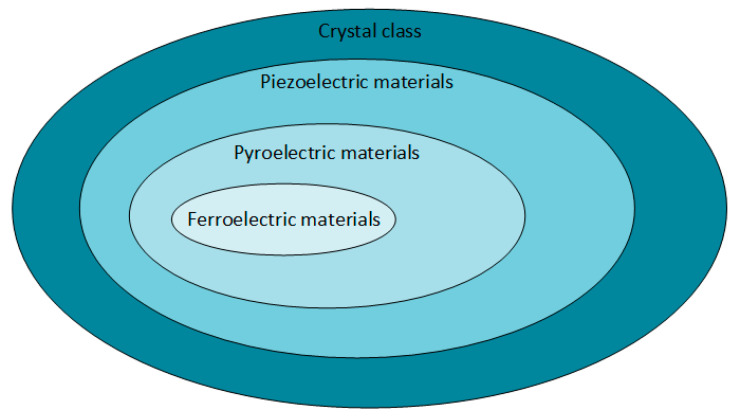
The relationship between piezo-, pyro-, and ferroelectric materials.

**Figure 5 sensors-20-03512-f005:**
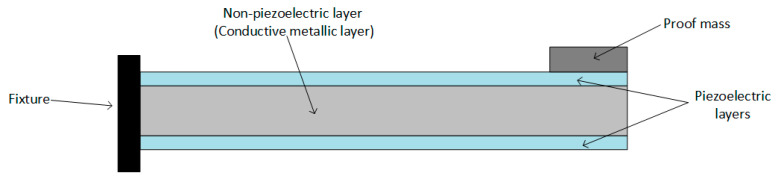
Cantilever beam transducer.

**Figure 6 sensors-20-03512-f006:**
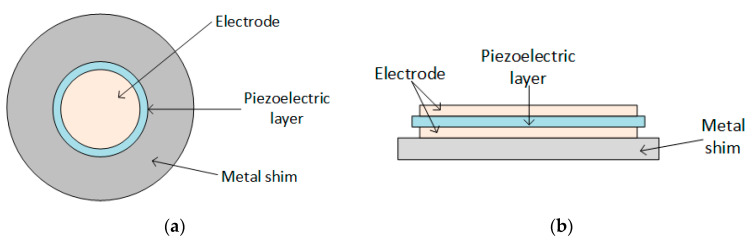
Circular diaphragm transducer: (**a**) Front view; (**b**) Side view.

**Figure 7 sensors-20-03512-f007:**
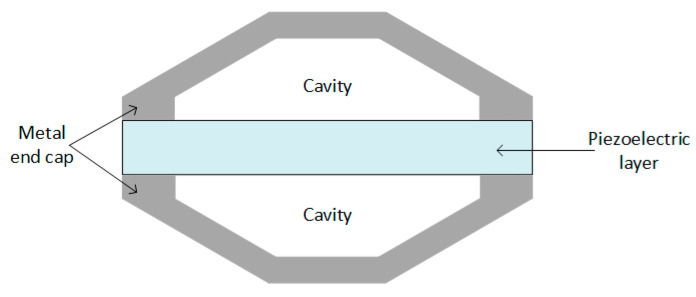
Cymbal transducer.

**Figure 8 sensors-20-03512-f008:**
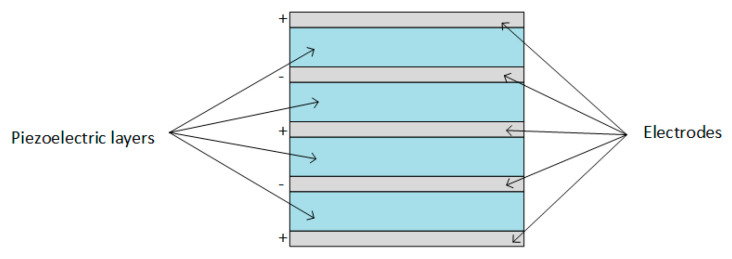
Stack piezoelectric transducer.

**Figure 9 sensors-20-03512-f009:**
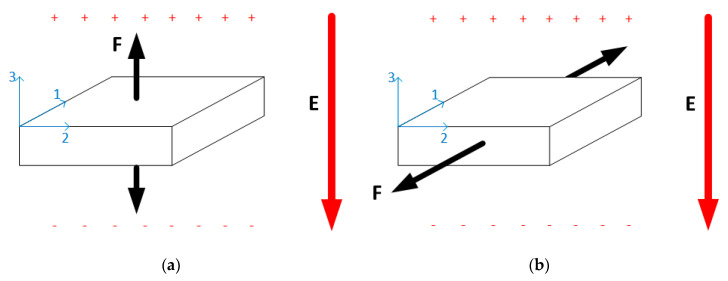
Working mode: (**a**) d_33_ mode (transverse mode) (**b**) d_31_ mode (longitudinal mode).

**Figure 10 sensors-20-03512-f010:**
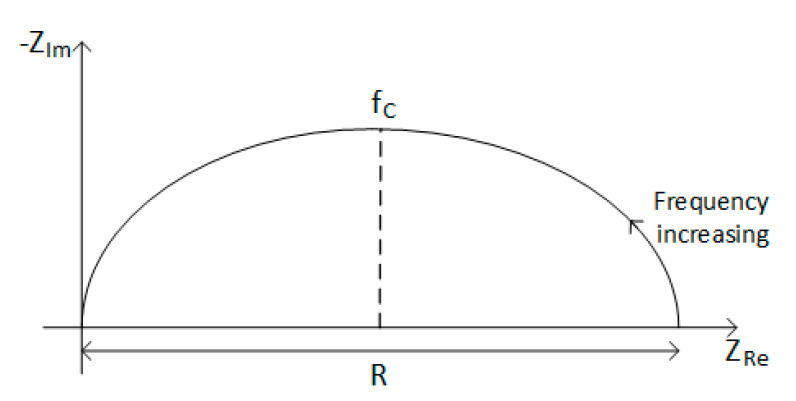
Nyquist plot of the ideal semi-circular response from a RC circuit.

**Figure 11 sensors-20-03512-f011:**
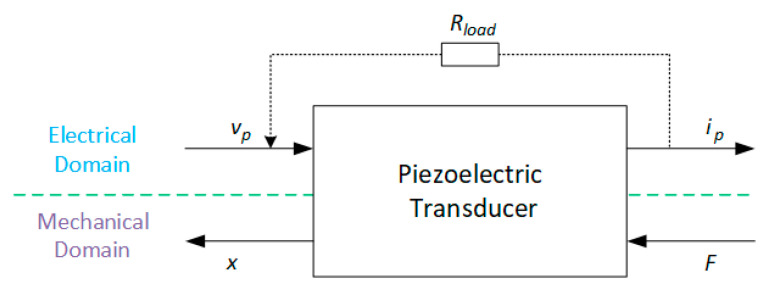
Two-port network model for the piezoelectric transducer.

**Figure 12 sensors-20-03512-f012:**
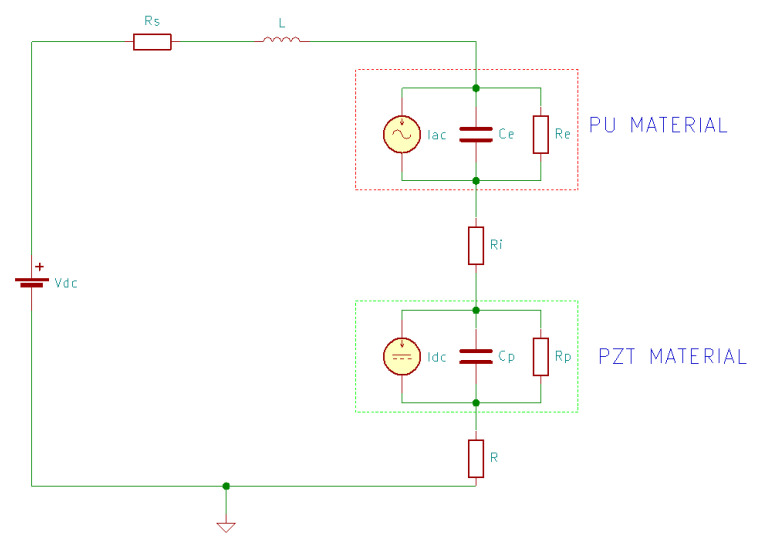
The equivalent circuit model [1].

**Figure 13 sensors-20-03512-f013:**
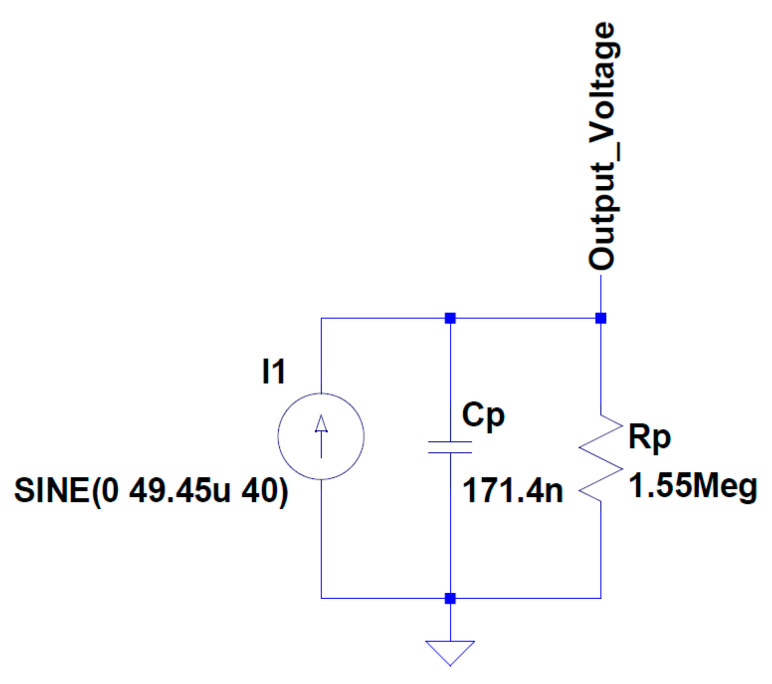
Piezoelectric SPICE model for QP20W transducer.

**Figure 14 sensors-20-03512-f014:**
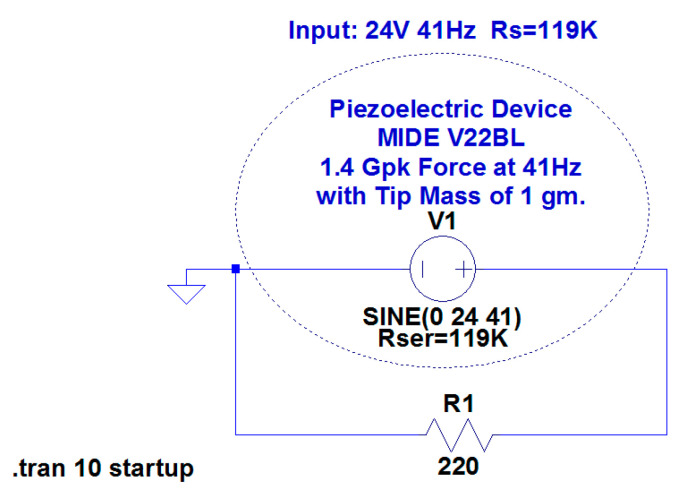
Piezoelectric SPICE model for MIDE V22BL transducer.

**Figure 15 sensors-20-03512-f015:**
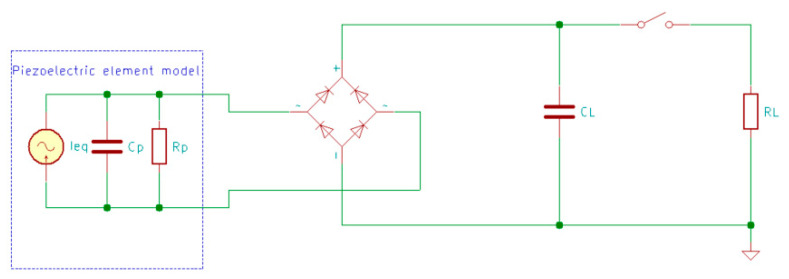
Standard AC-DC energy harvesting circuit with electric piezoelectric model.

**Figure 16 sensors-20-03512-f016:**
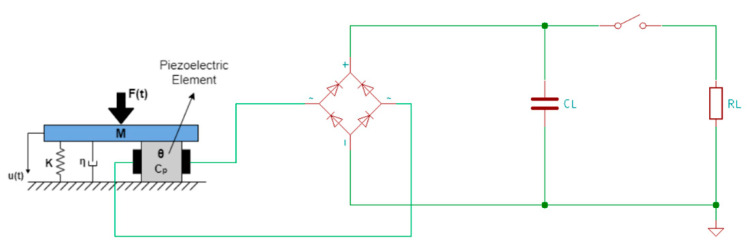
Standard AC-DC energy harvesting circuit with electromechanical piezoelectric model.

**Figure 17 sensors-20-03512-f017:**
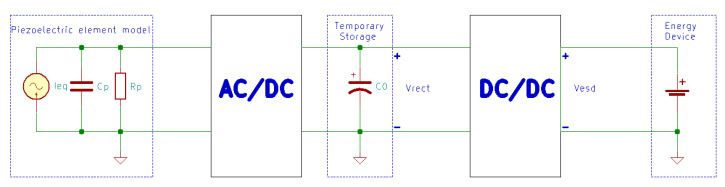
Generalized two-stage piezoelectric energy harvesting circuit.

**Figure 18 sensors-20-03512-f018:**
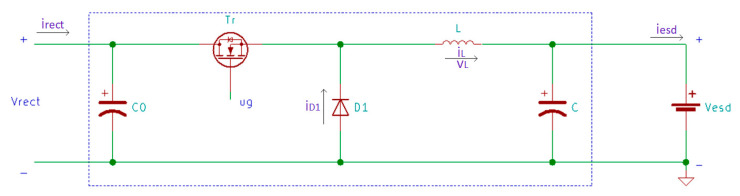
DC-DC converter in the two-stage energy harvesting scheme.

**Figure 19 sensors-20-03512-f019:**
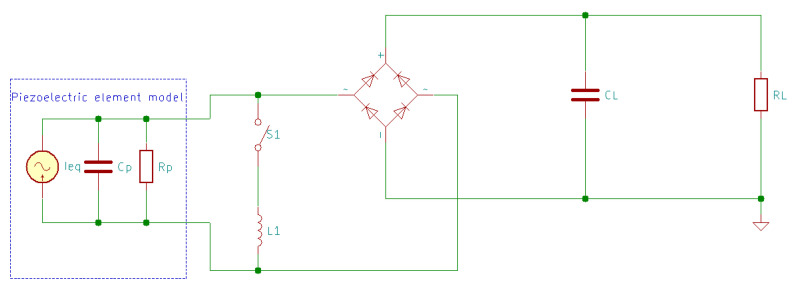
Parallel-Synchronized Switch Harvesting on Inductor (SSHI) energy harvesting circuit.

**Figure 20 sensors-20-03512-f020:**
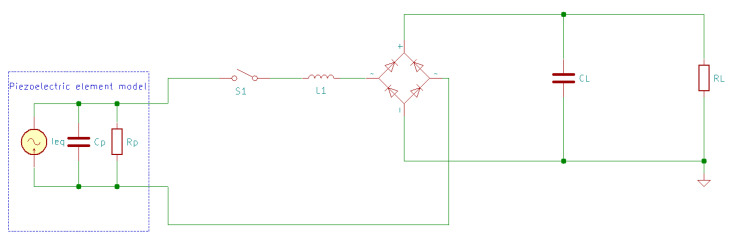
Series-SSHI energy harvesting circuit.

**Figure 21 sensors-20-03512-f021:**
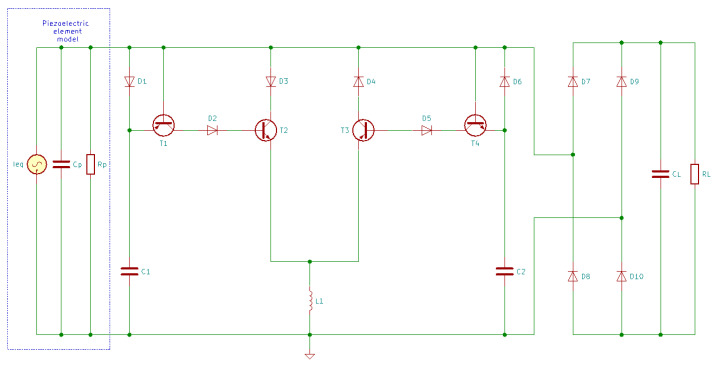
Self-powered parallel-SSHI energy harvesting circuit.

**Figure 22 sensors-20-03512-f022:**
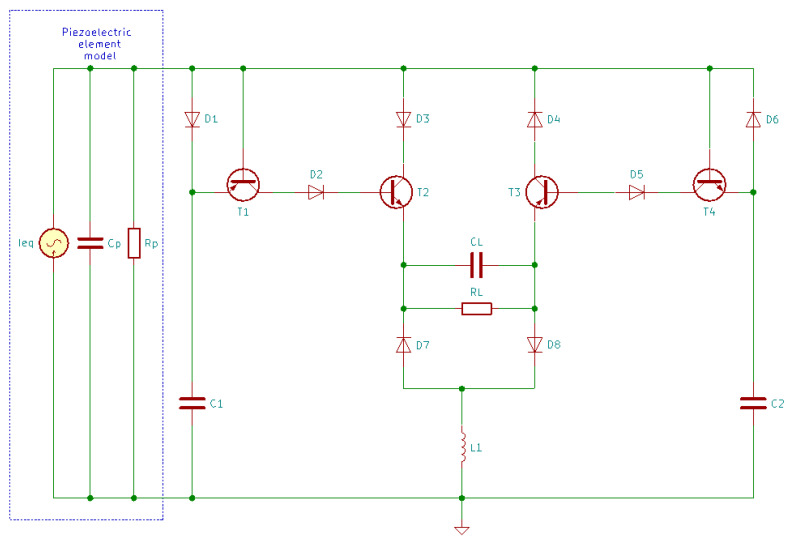
Self-powered series-SSHI energy harvesting circuit.

**Figure 23 sensors-20-03512-f023:**
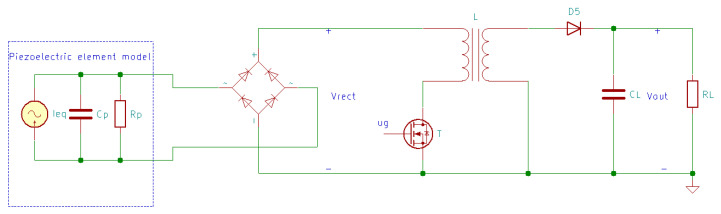
Synchronous Electrical Charge Extraction (SECE) energy harvesting circuit.

**Table 1 sensors-20-03512-t001:** Characteristics of piezoelectric material types [9].

Type	Description and Characteristics	Existing Solutions and Examples
Single crystal materials	Monocrystals vertically grown on a substrate with Bridgeman method or Flux method, etc.; Outstanding piezoelectric properties and are mostly used in sensors and actuators applications; Depending on the growing technique, can have different nanostructure forms.	Zinc-Oxide (ZnO); Lead Magnesium Niobate (or PMN) based nanostructures: PMN-PT.
Lead-based Piezoceramics	Polycrystalline materials with perovskite crystal structure; High piezoelectric effect and low dielectric loss; Simple fabrication process, compatible with MEMS fabrication; Highly toxic due to the presence of lead.	Most common are modified or doped PZT, such as: Lead Magnesium Niobate-PZT (PMN-PZT), PZT-5A, Zinc Oxide enhanced PZT (PZT-ZnO), etc.
Lead-free Piezoceramics	Non-toxic piezoceramics; Have lower transduction efficiency; Competitive lead-free materials are perovskite crystal structured type.	BaTiO_3_; Bismuth Sodium Titanate (BNT-BKT); Potassium Sodium Niobate (KNN) – based material: LS45, KNLNTS.
Piezopolymers	Electroactive Polymer (EAP); Flexible, non-toxic and light-weighted; Small electromechanical coupling than piezoceramics; Low manufacturing cost and rapid processing; Biocompatible, biodegradable, and low power consumption compared to other piezoelectric materials.	Can be used for piezo-MEMS fabrication; Polyvinylidene Fluoride (PVDS) derived polymers.

**Table 2 sensors-20-03512-t002:** Comparison between piezoelectric ceramics and piezoelectric polymers [63].

Properties/Parameters	Piezoelectric Ceramics (PZT)	Piezoelectric Polymers (PVDF)
Piezoelectricity	High	Low
Acoustic impedance (10^6^ kg m^−2^ s^−1^) ^1^	High (30)	Low (2.7)
Density (10^3^ kg m^−3^)	7.5	1.78
Relative permittivity (*ε/ε_0_*)	1200	12
Piezoelectric strain constant (10^−12^ C N^−1^)	*d_31_* = 110, *d_33_* = 225–590	*d_31_* = 23, *d_33_* = −33
Piezoelectric stress constant (10^−3^ V m N^−1^) ^1^	*g_31_* = 10, *g_33_* = 26	*g_31_*=216, *g_33_* = −330
Electromechanical coupling factor ^2^	*k_31_* = 30	*k_31_* = 12
Dielectric constant	1180	10–15
Mechanical flexibility ^1^	Poor	Outstanding
Curie temperature (°C)	386	80

^1^ Exceptional properties of PVDF for energy harvesting application vis-à-vis PZT ceramics. ^2^ % at 1 kHz.

**Table 3 sensors-20-03512-t003:** Properties of interest for various ceramics and single crystals [86].

Piezoelectric Material	$d_{31}(\mathrm{pm}/V)$	$s_{11}^{E}({\mathrm{pm}}^{2}/N)$	$\varepsilon_{33}^{T}/\varepsilon_{0}$	ρ (kg/m^3^)
PZT-5A	−171	16.4	1700	7750
PZT-5H	−274	16.5	3400	7500
PMN-PT (30% PT)	−921	52	7800	8040
PMN-PT (33% PT)	−1330	69	8200	8060
PMN-PZT	−2252	127	5000	7900
**Average**	**−989.6**	**56.2**	**5220**	**7850**

**Table 4 sensors-20-03512-t004:** Generated peak power for piezoelectric materials.

Piezoelectric Material	Peak Power (mW)	Volume	Frequency (Hz)
PVDF [87]	0.61	72 × 16 × 0.41 mm	2
PZT ceramic [88]	52	1.5 cm^3^	100
PZT fiber [89]	120	2.2 cm^3^	-
PMN-PZT single crystal [90]	0.015	20 × 5 × 0.5 mm	1744
PMN-PT single crystal [91]	3.7	25 × 5 × 1 mm	102

**Table 5 sensors-20-03512-t005:** Advantages and disadvantages of different configurations for piezoelectric transducers.

Type of Configuration	Features/Advantages	Disadvantages
Unimorph/Bimorph cantilever beam	Simple structure Low fabrication cost Lower resonance frequency Power output is proportional to proof mass High mechanical quality factor	Inability to resist a high impact force
Circular diaphragm	Compatible with pressure mode operation	Stiffer than a cantilever of the same size Higher resonance frequencies
Cymbal transducer	High energy output Withstands high impact force	Limited to applications demanding high-magnitude vibration sources
Stacked structures	Withstands high mechanical load Suitable for pressure mode operation Higher output from *d_33_* mode	High stiffness

**Table 6 sensors-20-03512-t006:** Summary of the most popular circuits used in piezoelectric energy harvesting.

Circuit	Description
AC-DC piezoelectric energy harvesting circuit	Rectifies the AC power generated by the piezoelectric transducers
Two stage piezoelectric energy harvesting circuit	Maximize the rectified power
SSHI technique	Uses a nonlinear technique to nullify the effect of the capacitive term and increases the output power
SECE technique	Compared with other techniques, it does not depend on the load

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
