# Peer review of "Piezoelectric Energy Harvesting Solutions: A Review"

_sensors, 2020, doi:10.3390/s20123512_

Round 1

Reviewer 1 Report

This is a good overview of piezoelectric harvester and the associated circuit, circuit models. Some additional information will add more value to this review paper.

Add a chart to compare mechanical Bandwidth of different piezoelectric harvester.

Add a table to summarize different power level over per unit volume of different piezoelectric material, frequency range.

Add a table to summarize all circuit techniques used to harvest piezoelectric harvesters discussed in Figure 21 through Figure 29.

High electric voltage and low current is a unique trait to piezoelectric harvesters. This will imply that low voltage CMOS process is not a good fit to be interface with such piezoelectric device as electrical overstress will break down the transistors. Please add this point to the literature.

Reviewer 2 Report

See attached pdf

Round 2

Reviewer 2 Report

No further comment. Article is ready for publication.